# Do Hospitalized Adult Patients with Acute Pharyngotonsillitis Need Empiric Antibiotics? The Impact on Antimicrobial Stewardship

**DOI:** 10.3390/microorganisms13030628

**Published:** 2025-03-10

**Authors:** Chih-Wei Liang, Mei-Cheng Hsiao, Shin-Huei Kuo, Shang-Yi Lin, Nai-Hwa Shih, Min-Han Hsieh, Tun-Chieh Chen, Po-Liang Lu

**Affiliations:** 1Division of Infectious Diseases, Department of Internal Medicine, Kaohsiung Medical University Hospital, Kaohsiung Medical University, Kaohsiung 807, Taiwan; eric483460@gmail.com (C.-W.L.); 1000420kmuh@gmail.com (S.-H.K.); amoe616@gmail.com (S.-Y.L.); idpaul@gmail.com (P.-L.L.); 2Department of Neurosurgery, Neurological Institute, Taichung Veterans General Hospital, Taichung 407, Taiwan; iaaron10251@gmail.com; 3Department of Laboratory Medicine, Kaohsiung Medical University Hospital, Kaohsiung Medical University, Kaohsiung 807, Taiwan; imm8556@yahoo.com.tw; 4College of Medicine, Kaohsiung Medical University, Kaohsiung 807, Taiwan; 5Department of Internal Medicine, Kaohsiung Municipal Min-Sheng Hospital, Kaohsiung 802, Taiwan; 970388kmuh@gmail.com; 6Center for Medical Education and Humanizing Health Professional Education and Center for Tropical Medicine and Infectious Disease Research, Kaohsiung Medical University, Kaohsiung 807, Taiwan; 7Center for Liquid Biopsy and Cohort Research, Kaohsiung Medical University, Kaohsiung 807, Taiwan

**Keywords:** acute pharyngotonsillitis, group A streptococcus, modified Centor criteria, antimicrobial stewardship

## Abstract

Acute pharyngotonsillitis is a common reason to visit primary care providers. Group A Streptococcal (GAS) pharyngitis is the most common bacterial infection which needs antibiotic treatment. GAS accounts for only 10–15% of adult acute pharyngitis cases. The overuse of antibiotics for viral pharyngotonsillitis is common and may lead to inappropriate antimicrobial stewardship and the emergence of bacterial resistance. However, the etiology of acute pharyngotonsillitis for hospitalized adult patients is rarely studied. So, we reported the 10-year surveillance data of hospitalized adult patients with acute pharyngotonsillitis in a regional hospital in Taiwan. Every consecutive adult patient admitted with acute pharyngotonsillitis in 2011–2021 was recruited for a complete etiology study. The etiology of acute pharyngotonsillitis was identified in 117 patients. Overall, 42 herpes simplex virus cases, 26 adenovirus cases, 16 acute human immunodeficiency virus cases, 12 influenza cases, three parainfluenza cases, six Epstein–Barr virus cases, one cytomegalovirus case, four enterovirus cases, one varicella-zoster virus case, four *Mycoplasma pneumoniae* cases, one *Chlamydophila pneumoniae* case, and only one GAS case were identified. The average of the points for the Modified Centor Criteria was 1.38 (55% of patients with 0–1 points and 45% with 2–3 points). However, 88.9%of patients received antibiotics at the emergency department, and 76.9%also received antibiotics while hospitalized. Only a few patients required antibiotic treatment, while the majority of patients with viral illness needed only symptomatic treatment. However, distinguishing viral etiology from GAS pharyngitis is challenging even in the presence of tonsil exudates, high C-reactive protein, and leukocytosis. A diagnostic algorithm and the application of the Modified Centor Criteria should be considered for hospitalized adults with acute pharyngotonsillitis to improve antimicrobial stewardship.

## 1. Introduction

Acute pharyngotonsillitis is one of the common reasons for presenting to primary care providers for both adults and children. It ranges from mild, self-limited viral illness to severe, life-threatening bacterial infection, such as Ludwig Angina or Lemierre’s syndrome. Group A Streptococcal (GAS) pharyngitis is the most common bacterial infection which needs antibiotic treatment. In previous studies, GAS accounted for only 10–15% of acute pharyngitis cases in adults. However, 66–78% of adults with pharyngitis received antibiotic treatment [1,2,3,4]. According to epidemiological data from the National Health Insurance Research Database in Taiwan, the antibiotic prescription rate for acute tonsillitis was 16.8% between 2000 and 2009 (28.4% in 2000 and 10.9% in 2009, respectively). More antibiotics were used by patients of older age, in eastern Taiwan, and treated at medical centers or by non-pediatricians and ENT doctors [5]. As many as one-third to one-half of adult acute respiratory tract infections may be inappropriately treated with antibiotics [6,7,8,9]. The overuse of antibiotics for viral pharyngotonsillitis may lead to inappropriate antimicrobial stewardship and the emergence of bacterial resistance [10,11]. The etiology of acute pharyngotonsillitis for hospitalized adult patients is rarely studied [12,13]. The Modified Centor Criteria or McIsaac Score (tonsil exudates, tender anterior cervical adenopathy, high fever, and the absence of cough, adjusted by age 3–14: +1; ≥45: −1) can guide antibiotic use for GAS pharyngitis. It is suggested that patients with >2–3 Modified Centor Criteria points receive empiric antibiotics and undergo further testing, such as the GAS rapid test and throat swab bacterial culture. Patients with only 0–1 points are said to have a viral etiology, for which no empiric antibiotic treatment is favored [14,15].

Therefore, we reported the 10-year surveillance data of hospitalized adult patients with acute pharyngotonsillitis in a community hospital in Taiwan. We also evaluated the impact of the application of the Modified Centor Criteria on antimicrobial stewardship.

## 2. Materials and Methods

### 2.1. Study Populations and Laboratory Testing

This study was conducted at a 420-bed regional hospital in southern Taiwan. Acute pharyngotonsillitis was defined as the rapid onset of a sore throat and pharyngeal inflammation (with or without exudative swollen tonsil), although there is no definitive consensus on admission criteria for adult patients with acute pharyngotonsillitis. If the patient had prolonged fever and made repeated visits to our emergency department or high fever (>39 °C) and acute illness with severe odynophagia and dysphagia (difficulty swallowing) or leukocytosis (>10,000/mm^3^; normal range: 3920–9240/mm^3^), and a high C-reactive protein level (>50 mg/L; normal range: ≦5 mg/L), the patient would be admitted to the medical ward via the emergency department at our hospital. Every consecutive adult patient (more than 18 years old) admitted with acute pharyngotonsillitis between 2011 and 2021 was recruited for a complete etiology study, including a throat swab for viral isolation and bacterial culture, Group A Streptococcus antigen rapid test (Abbott, Lake Forest, IL, USA), viral antigen test for influenza, parainfluenza, herpes simplex virus (HSV), adenovirus, and respiratory syncytial virus (Denka, Tokyo, Japan), and serology for Chlamydophila pneumoniae, Mycoplasma pneumoniae (Biocard^TM^ Labsystems Diagnostics, Vantaa, Finland), adenovirus, HSV, varicella-zoster virus (VZV), Epstein–Barr virus (EBV) (NOVATec Immundiagnostica GmbH, Dietzenbach, Germany), cytomegalovirus (CMV), human immunodeficiency virus (HIV) (Architect Abbott, Lake Forest, IL, USA), and influenza A & B rapid RNA test (ID NOW^TM^ Abbott, Lake Forest, IL, USA). The laboratory procedures were performed according to the manufacturers’ instructions. The influenza rapid antigen test or rapid RNA test is a routine test for patients at emergency departments presenting with high fever and upper respiratory symptoms (such as sore throat and cough). This kind of patient will not be admitted with the diagnosis of acute pharyngotonsillitis. Patients with a positive influenza rapid test were excluded from our cohort because these patients would be treated with an anti-influenza virus agent first and maybe with additional antibiotics for superimposed bacterial pneumonia, which was not considered inappropriate antimicrobial stewardship. A chart review was conducted to collect demographic information, including medical history, clinical symptoms, antibiotic use, and laboratory data. This study was approved by the Institutional Review Board of Kaohsiung Medical University Hospital (KMUHIRB-E(II)-20240114).

### 2.2. Statistical Analysis

All data were analyzed using SPSS version 20.0 (IBM Corp, Armonk, NY, USA). Continuous variables among different causal pathogens were analyzed with a non-parametric test, namely, the Kruskal–Wallis test. Categorical variables were compared using the chi-square test. When more than 20% of cells had expected frequencies <5, Fisher’s exact test was applied. A *p*-value of < 0.05 was considered statistically significant (two-tailed).

## 3. Results

### 3.1. The Etiologies of Hospitalized Adult Patients with Acute Pharyngotonsillitis

After excluding 338 patients with positive results for the influenza rapid test, including 221 positive rapid antigen and 64 positive RNA tests of type A influenza, and 38 positive rapid antigen and 15 positive RNA tests of type B influenza, a total 290 patients with acute pharyngotonsillitis were admitted to the medical ward in 2011–2021. The etiology of acute pharyngotonsillitis was identified in 117 patients. No duplicated viral–viral or viral–bacterial co-infections were identified. For the other 173 patients, we were unable to identify their etiology by current methods (Figure 1). The white blood cell (WBC) counts and C-reactive protein (CRP) levels were not significantly different between the 117 patients with identified etiology and 173 patients without identified etiologies. In all, 42 HSV cases, 26 adenovirus cases, 16 acute HIV cases, 12 influenza cases, 3 parainfluenza cases, 6 EBV cases, 1 CMV case, 4 enterovirus cases, 1 VZV case, 4 *Mycoplasma pneumoniae* cases, 1 *Chlamydophila pneumoniae* case, and 1 Group A Streptococci case were identified.

### 3.2. The Demographic Characteristics and Clinical Symptoms of Hospitalized Patients with Acute Pharyngotonsillitis Across Different Causal Pathogens

The average ages of patients with influenza (41.3 years) and parainfluenza (59.7 years) were significantly higher than of those with other causal pathogens. Patients with influenza had a significantly higher incidence of cough than others. Neck lymphadenopathy was significantly more common in patients with EBV, CMV (42.9%), and acute HIV (31.3%). Patients with acute HIV infection had a significantly higher likelihood of manifesting nausea/diarrhea, while those with CMV/EBV commonly had splenomegaly (Table 1).

### 3.3. The Laboratory Data of Hospitalized Patients with Acute Pharyngotonsillitis Patients Across Different Causal Pathogens

The patients with HSV and adenovirus infections tended to have leukocytosis, neutrophilia, and high CRP, making differentiation from bacterial infection difficult. Patients with acute HIV tended to have lower WBC and neutrophil counts and more atypical lymphocytes. Patients with CMV/EBV (85.7%) and acute HIV infection (50%) had a significantly higher incidence of hepatitis (glutamic pyruvic transaminase [GPT] > 2 times the upper limit of the normal range) (Table 2).

### 3.4. The Distribution of Modified Centor Scores and Administration of Antibiotics Among Different Causal Pathogens

The average of the points for the Modified Centor Criteria was 1.38 (55% of patients with 0–1 points, 32.5% with 2 points, and 12.8% with 3 points). No patients had more than 4 points, the threshold for empiric antibiotic treatment. However, 88.9% of patients received antibiotics at the emergency department, and 76.9% received antibiotics while hospitalized. Patients with HSV pharyngotonsillitis had a significantly higher chance of receiving antibiotics, while those with mononucleosis-like syndrome, such as EBV, CMV, and acute HIV infection, had significantly lower rates of antibiotic prescriptions from infectious disease physicians (Table 3).

## 4. Discussion

In our study, 95% of hospitalized adults with acute pharyngotonsillitis had a viral etiology except for four cases of *Mycoplasma pneumoniae*, one case of Group A Streptococcus, and one case of *Chlamydophila pneumoniae*. More than half (55%) of these patients with pharyngotonsillitis had 0–1 points on the Modified Centor Criteria. However, 88.9% of patients received antibiotics in the emergency department, and 76.9% received antibiotics while hospitalized. Epidemiological studies of pediatric pharyngotonsillitis have been common but surveillance in adult patients has seldom been reported. The surveillance of the etiology of acute respiratory infections by viral culture and polymerase chain reaction (PCR) during the influenza season in an acute ambulatory care clinic or emergency department in San Francisco in 2002 identified influenza in 20% of patients, rhinovirus in 10%, and respiratory syncytial virus (RSV) in 5%, followed by human metapneumovirus (HMPV), human coronavirus, parainfluenza virus, and adenovirus. In 60% of patients, no pathogens were identified. Human metapneumovirus, human coronavirus, and rhinovirus were only identified by PCR, and they were difficult to isolate by traditional viral culture [12]. Two reports using PCR-based tools for adult patients with pharyngitis showed that the most commonly identified viruses were rhinovirus, coronavirus, and influenza virus, followed by HMPV, RSV, parainfluenza virus, adenovirus, and enterovirus. In these reports, 42% and 29% of patients showed negative results, respectively [16,17]. Two pediatric reports in Taiwan showed that adenovirus, influenza, enterovirus, and HSV were the primary causal pathogens in hospitalized children with pharyngitis and tonsillitis. RSV and enterovirus were more common in younger children <1–3 years, while HSV, adenovirus, and influenza were more common in preschool- and school-age children [3,18]. The difference among these surveillance studies depended on whether the setting was inpatient or outpatient, whether the focus was on pharyngitis/tonsillitis or other acute respiratory illnesses, the age groups studied, and whether culture-based or PCR-based methods were used, even with viral antibody serology tests.

Different causal pathogens’ characteristics can help physicians identify the possible etiology of acute pharyngotonsillitis. Patients with influenza were older and had a higher incidence of cough, presenting with the sudden onset of high fever, headache, and myalgia. Patients with HSV, adenovirus, and enterovirus infection had higher rates of leukocytosis and neutrophilia and high CRP, resembling acute bacterial infections. Tonsil exudates are often present in HSV, EBV/CMV, adenovirus, and enterovirus infections. Enterovirus and HSV infections were more likely to present vesicles and ulcers in the palate and pharyngeal wall. Adenovirus infections often presented with conjunctivitis. Fever lasting more than one week was frequently encountered in patients with HSV, EBV/CMV, adenovirus, and acute HIV. Mononucleosis-like syndrome in EBV, CMV, and acute HIV infection was noted in younger patients with cervical lymphadenopathy, hepatitis, and splenomegaly. Skin rashes and body weight loss were sometimes present in patients with acute HIV infection (Table 4). Acute HIV infection should also be considered in the differential diagnosis of acute or recurrent pharyngotonsillitis [19]. Our findings are consistent with a multicenter study of acute HIV infection in Taiwan. The most common symptoms were fever, fatigue, myalgia, diarrhea, and headache. The most common physical findings were pharyngitis, skin rash, lymphadenopathy, oral ulcer, aseptic meningitis, and genital ulcer. We should not neglect the diagnosis of acute HIV infection in male youth with pharyngotonsillitis who presented with prolonged fever, lymphadenopathy, splenomegaly, skin rash, headache, aseptic meningitis, gastrointestinal symptoms, and body weight loss [20].

The application of clinical prediction rules, such as the Centor Criteria or FeverPAIN Score, to predict Group A Streptococcal pharyngitis remains controversial. One randomized controlled trial demonstrated that the area under the receiver operating characteristics curve (AUROC) was 0.62 for the Centor Criteria and 0.59 for the FeverPAIN Score [21]. One large-scale retrospective study showed that the Centor score yielded an AUROC of 0.72 [22]. Another clinical observational study found 100% sensitivity and 68.7% specificity, giving a positive predictive value of 12.7% and a negative predictive value of 100% [23]. An updated systematic review and meta-analysis revealed that the Modified Centor Criteria at thresholds of 2 and 3 had high sensitivities (both over 0.75) and high false positive rates (both over 0.5) [14]. Although the positive predictive value of the Modified Centor Criteria for GAS pharyngitis is fair and related to the prevalence of GAS, a low score (0–1 points) can exclude the possibility of GAS infection and reduce unnecessary antibiotic use [24,25]. Although the diagnostic accuracy for GAS of the Modified Centor Criteria is not good, a low score (0–1 points) had nearly 90% negative predictive value in most studies. In our cohort, if patients with only 0–1 points for the Modified Centor Criteria were not administered empiric antibiotics, we could eliminate 61.5% of unnecessary antibiotic use (64 patients with 0–1 points/initial 104 patients with empiric antibiotics).

In current clinical practice, inflammatory markers, such as C-reactive protein or procalcitonin (PCT), are often used to guide antibiotic prescription. Earlier reports indicated higher WBC counts and CRP levels in GAS tonsillitis [13]. However, more recent studies showed that CRP levels do not correlate well with GAS pharyngitis [26,27]. Although our study had only one case of GAS infection, HSV, adenovirus, and enterovirus infections also had high CRP levels, with 26.1–41.5% of these cases having a CRP value higher than 100 mg/L. Thus, CRP is not practical for identifying patients who require antibiotic therapy. Procalcitonin is not widely used in patients with pharyngotonsillitis, and its role in the differential diagnosis of the etiology of acute respiratory illness remains unclear.

Current guidelines vary in the use of the rapid antigen detection test (RADT) for GAS in patients with acute pharyngotonsillitis. Spanish, French, and Infectious Disease Society of America guidelines suggest that all children older than three years with acute pharyngitis should be tested for GAS infection, regardless of their clinical scores. The World Health Organization and the Canadian, European, and American Academy of Pediatrics have recommended using clinical scoring systems to identify patients for testing. For example, the RADT should be performed on patients with 2–3 Modified Centor Criteria points. The throat culture is not recommended for the routine primary evaluation of adults with pharyngitis. Immediate antibiotic treatment should be provided for patients with positive RADT results or high clinical scores, such as 3–4 Modified Centor Criteria points [28,29,30].

This study had some limitations. First, we could only use routine laboratory tests in a single community hospital to identify 117 hospitalized adults with acute pharyngotonsillitis, leaving 173 (60%) patients with unidentified etiology. We conducted a sensitivity analysis to compare WBC counts and CRP levels, which showed no significant difference between the identified etiology and unidentified groups. Multiplex nucleic acid amplification techniques, such as the FilmArray^®^ test, could potentially identify more etiologies and increase diagnostic accuracy and detection sensitivity [31]. However, these multiplex PCR assays are too expensive for routine clinical practice and were not widely available ten years ago. If these point-of-care nucleic acid amplification tests become widely available in our clinical setting, we can reduce reliance on presumptive antibiotic treatment and thereby facilitate antimicrobial stewardship [32,33]. Second, not all patients received paired serology tests for viral infections in the acute and convalescent phases. Some patients did not return to our outpatient service after discharge because their symptoms had completely resolved, resulting in missed etiologies that could have been identified by convalescent serology tests.

## 5. Conclusions

In conclusion, the primary cause of pharyngotonsillitis in hospitalized adult patients was viral infection. Most patients received unnecessary antibiotics, which may be associated with adverse events and lead to bacterial resistance. C-reactive protein and WBC counts were unreliable in distinguishing bacterial or viral infections. The Modified Centor Criteria could be considered the first step in determining empiric antibiotic prescriptions for adult patients with pharyngotonsillitis, which could potentially reduce unnecessary antibiotic use by half and facilitate antimicrobial stewardship. If point-of-care nucleic acid amplification tests become widely available in clinical practice, the etiological diagnosis of acute pharyngotonsillitis and antibiotic use would be more precise.

## Figures and Tables

**Figure 1 microorganisms-13-00628-f001:**
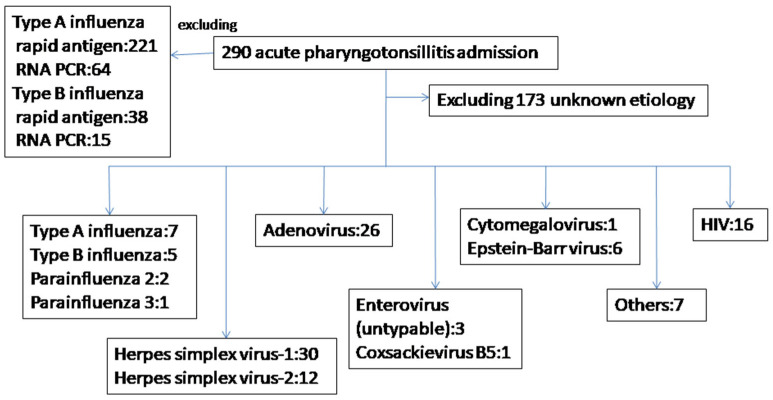
The etiologies of hospitalized adult patients with acute pharyngotonsillitis.

**Table 1 microorganisms-13-00628-t001:** The demographic characteristics and clinical symptoms of hospitalized patients with acute pharyngotonsillitis across different causal pathogens.

Etiology	Influenza A (7) Influenza B (5)	Parainfluenza 2 (2) Parainfluenza 3 (1)	HSV-1 (30) HSV-2 (12)	EBV (6) CMV (1)	ADV (26)	Enterovirus (4)	Acute HIV (16)	*Myacoplasma pneumoniae*(4) GAS (1) VZV (1) *Chlamydophila pneumoniae*(1)
Age (mean ± SD)	41.3 ± 12.6 *	59.7 ± 17.6 *	31.5 ± 15.8	26.9 ± 11.3	31.1 ± 10.3	31.3 ± 10.5	28.9 ± 5.9 *	32.3 ± 9.6
Gender (M/F)	7/5	2/1	24/18	4/3	9/17 *	3/1	16/0	6/1
Diagnosed by viral isolation	12	12	35	0	25	4	0	0
Antigen	0	0	2	0	1	0	0	1
Antibody	0	0	9	7	0	0	16	3
Initial BT	38.3 ± 0.9	37.7 ± 0.9	38.2 ± 1.0	37.8 ± 1.1	38.4 ± 1.0	36.9 ± 0.6 *	38.0 ± 1.2	38.3 ± 1.1
Highest BT	38.3 ± 0.8	37.7 ± 0.9	38.7 ± 0.9	38.3 ± 0.6	38.9 ± 1.0	38.8 ± 0.9	38.3 ± 1.3	39.0 ± 1.1
Initial HR	100.2 ± 16.1	104.7 ± 32.6	104.4 ± 19.3	109.8 ± 17.9	106.3 ± 12.5	78.3 ± 19.5 *	101.3 ± 20.0	106.3 ± 12.4
Highest HR	100.2 ± 16.1	104.7 ± 32.6	108.7 ± 16.0	110.8 ± 17.3	107.8 ± 10.8	84.5 ± 22.2 *	102.0 ± 20.3	108.9 ± 9.9
Lowest HR	64.8 ± 6.8	82 *	64.1 ± 9.3	70.0 ± 12.5	63.6 ± 8.3	55.0 ± 9.5	61.0 ± 5.3	62.7 ± 8.6
Fever	12 (100%)	0	41 (97.6%)	7 (100%)	26 (100%)	1 (25.0%)	16 (100%)	6 (85.7%)
Sore throat	10 (83.3%)	1 (33.3%)	38 (90.55)	7 (100%)	23 (88.5%)	1 (25.0%)	14 (87.5%)	6 (85.7%)
Headache	3 (25%)	0	16 (38.1%)	1 (14.3%)	6 (23.1%)	0	6 (37.5%)	1 (14.3%)
Myalgia	4 (33.3%)	2 (66.7%)	10 (23.8%)	4 (57.1%)	8 (30.8%)	0	6 (37.5%)	0
Arthralgia	3 (25%) *	1 (33.3%)	4 (9.5%)	1 (14.3%)	0	0	0	0
Cough	12 (100%) *	0	17 (40.5%)	3 (42.9%)	12 (46.2%)	0	7 (43.8%)	2 (28.6%)
Neck LAP	0	0	2 (4.8%)	3 (42.9%) *	1 (3.8%)	0	5 (31.3%) *	0
Vesicles/ulcers	1 (8.3%)	0	8 (19%)	0	2 (7.7%)	1 (25.0%)	2 (12.5%)	0
Tonsil exudate	0	0	8 (19%)	3 (42.9%) #1	4 (15.4%)	1 (25.0%)	3 (18.8%)	0
Conjunctivitis	1 (8.3%)	0	0	0	2 (7.7%)	0	0	0
Rhinorrhea	5 (41.7%) *	0	5 (11.9%)	0	4 (15.4%)	1 (25.0%)	0	0
Nasal congestion	9 (8.3%)	0	4 (9.5%)	1 (14.3%)	1 (3.8%)	1 (25.0%)	0	2 (28.6%)
Hoarseness	1 (8.3%)	0	1 (2.4%)	0	0	0	0	0
Rash	1 (8.3%)	0	1 (2.4%)	1 (14.3%)	0	0	3 (18.8%) #2	1 (14.3%)
Nausea/vomiting	1 (8.3%)	1 (33.3%)	6 (14.3%)	1 (14.3%)	5 (19.2%)	1 (25.0%)	7 (43.8%) *	2 (28.6%)
Diarrhea	0	0	7 (16.7%)	0	5 (19.2%)	2 (50.0%)	7 (43.8%) *	1 (14.3%)
Splenomegaly	0	0	1 (2.4%)	4 (57.1%) *	0	0	2 (12.5%)	2 (28.6%) #3
BWL	0	0	0	0	0	0	1 (6.3%)	0

#1 *p* = 0.084: tonsil exudates in mononucleosis-like syndrome caused by EBV or CMV. #2 *p* =0.053: rash in acute HIV. #3 *p* =0.091: splenomegaly in others. * Star symbol indicates *p* < 0.05 according to non-parametric test and chi-square test. Abbreviations: HSV—herpes simplex virus; EBV—Epstein–Barr virus; CMV—cytomegalovirus; ADV—adenovirus; HIV—human immunodeficiency virus; GAS—Group A Streptoccoci; VZV—varicella-zoster virus; BT—body temperature; HR—heart rate; LAP—lymphadenopathy; BWL—body weight loss.

**Table 2 microorganisms-13-00628-t002:** The laboratory data of hospitalized patients with acute pharyngotonsillitis across different causal pathogens.

Etiology	Influenza A (7) Influenza B (5)	Parainfluenza 2 (2) Parainfluenza 3 (1)	HSV-1 (30) HSV-2 (12)	EBV (6) CMV (1)	ADV (26)	Enterovirus (4)	Acute HIV (16)	*Myacoplasma pneumoniae* (4) GAS (1) VZV (1) *Chlamydophila pneumoniae* (1)
Initial WBC (/mm^3^)	7601 ± 3884	5467 ± 2295	11,902 ± 5580 *	11,246 ± 3743	11,958 ± 5322	11,045 ± 4076	5479 ± 1808 *	11,224 ± 3702
Neutrophils %	67.8 ± 21.0	53.9 ± 21.9	77.7 ± 14.6 *	52.6 ± 16.2 *	77.3 ± 17.3 *	78.7 ± 10.4	63.6 ± 11.9 *	79.4 ± 7.6
Lymphocytes %	21.6 ± 16.2	35.2 ± 20.0 *	12.0 ± 5.5 *	35.6 ± 18.8 *	11.4 ± 3.8 *	15.6 ± 10.4	25.7 ± 10.2 *	13.3 ± 6.1
Monocytes %	8.9 ± 3.7	8.1 ± 0.3 *	7.5 ± 2.9	4.6 ± 1.9 *	7.1 ± 3.0	5.5 ± 0.9	7.6 ± 4.1	6.8 ± 2.5
Atypical lymphocytes %	0.1 ± 0.3	0 ± 0	0.2 ± 0.6	2.9 ± 2.7 *	0 ± 0.2 *	0 ± 0	1.2 ± 1.8	0 ± 0
Platelets(X1000) (/mm^3^)	197 ± 56	154 ± 69	219 ± 68	191 ± 53	248 ± 87 *	202 ± 49	159 ± 62 *	199 ± 56
Highest WBC (/mm^3^)	7984 ± 3733 *	5663 ± 1954	12,499 ± 5327 *	11,460 ± 3878	11,958 ± 5322	11,045 ± 4076	6388 ± 1639 *	11,751 ± 2767
Lowest WBC (/mm^3^)	3071 ± 848 *	3220 ± 566	7948 ± 2041 *	8682 ± 3645	6714 ± 3338	11,790	4286 ± 1726 *	5988 ± 1590
Highest PLTs(X1000) (/mm^3^)	200 ± 54	158 ± 63	252 ± 90	209 ± 53	259 ± 86	217 ± 56	203 ± 62	212 ± 50
Lowest PLTs(X1000) (/mm^3^)	155 ± 45	114 ± 31	203 ± 64	180 ± 58	215 ± 94	174 ± 35	135 ± 53 *	184 ± 62
Highest lymphocytes %	28.8 ± 17.6	36.3 ± 18.3	17.9 ± 9.7 *	41.5 ± 24.7	16.4 ± 8.7 *	15.9 ± 10.1	37.0 ± 12.1 *	26.1 ± 19.2
Highest monocytes %	9.3 ± 3.4	8.3 ± 0.5	8.2 ± 3.1	7.0 ± 2.3	7.5 ± 3.1	5.5 ± 0.9	9.7 ± 4.0 *	6.3 ± 1.4
Highest atypical lymphocytes %	0.1 ± 0.3 *	0 ± 0	0.9 ± 1.6	4.4 ± 5.7	0.2 ± 0.6 *	0 ± 0	2.3 ± 2.3 *	0.3 ± 0.8
WBC > 10,000/mm^3^	4 (33.3%)	0	31 (73.8%) *	4 (57.1%)	15 (57.7%)	3 (75.0%)	0	4 (57.1%)
Initial GOT (IU/L)	37.6 ± 22.4	52.0 ± 29.7	28.9 ± 16.4	118.2 ± 65.6	38.7 ± 27.0	20	185.7 ± 324.9	15.0
GPT (IU/L)	51.5 ± 51.6	51.3 ± 58.0	23.3 ± 13.9 *	140.6 ± 90.7	27.0 ± 20.0	24.0 ± 11.3	204.2 ± 447.7	65.4 ± 59.2
Highest GOT (IU/L)	37.6 ± 22.4	52.0 ± 29.7	29.3 ± 16.1	104.6 ± 68.3	29.6 ± 11.7	20	177.5 ± 279.7	59.0 ± 62.2
GPT (IU/L)	51.5 ± 51.6	51.3 ± 58.0	25.2 ± 18.8 *	140.6 ± 90.7	36.3 ± 50.0	24.0 ± 11.3	220.6 ± 443.1	76.4 ± 67.3
GPT > 2ULN	2 (18.2%)	1 (33.3%)	1 (2.6%) *	6 (85.7%) *	3 (11.5%)	0	8 (50.0%) *	3 (42.9%)
Initial CRP (mg/L)	35.3 ± 42.9	9.1 ± 8.7	86.6 ± 60.7 *	23.9 ± 27.1	73.4 ± 48.3	106.0 ± 77.8	35.3 ± 22.2 *	70.8 ± 68.7
Highest CRP (mg/L)	61.6 ± 70.7	9.1 ± 8.7	92.7 ± 59.0 *	24.4 ± 26.7 *	83.6 ± 64.7	106.0 ± 77.8	37.6 ± 20.4 *	71.5 ± 68.0
CRP > 100 mg/L	2 (20%)	0	17 (41.5%) *	0	6 (26.1%)	1 (33.3%)	0 *	2 (28.6)

* Star symbol indicates *p* < 0.05 according to non-parametric test and chi-square test. Abbreviations: HSV—herpes simplex virus; EBV—Epstein–Barr virus; CMV—cytomegalovirus; ADV—adenovirus; HIV—human immunodeficiency virus; GAS—Group A Streptoccoci; VZV—varicella-zoster virus; WBC—white blood cell; PLTs—platelets; GOT—glutamic oxaloacetic transaminase; GPT—glutamic pyruvic transaminase; CRP—C-reactive protein; ULN—upper limit of normal range.

**Table 3 microorganisms-13-00628-t003:** The distribution of modified Centor scores and administration of antibiotics among different causal pathogens.

Etiology	Influenza A (7) Influenza B (5)	Parainfluenza 2 (2) Parainfluenza 3 (1)	HSV-1 (30) HSV-2 (12)	EBV (6) CMV (1)	ADV (26)	Enterovirus (4)	Acute HIV (16)	*Myacoplasma pneumoniae* (4) GAS (1) VZV (1) *Chlamydophila pneumoniae* (1)
Modified Centor score	0.42 ± 0.51 *	0.67 ± 1.15	1.45 ± 0.92	1.86 ± 1.07	1.42 ± 0.95	2.00 ± 0.82	1.63 ± 0.89	1.43 ± 0.79
Score: 0	7 (58.3%)	2 (66.7%)	7 (16.7%)	0	4 (15.4%)	0	2 (12.5%)	1 (14.3%)
Score: 1	5 (41.7%)	0	14 (33.3%)	4 (57.1%)	11 (42.3%)	1 (25.0%)	4 (25.0%)	2 (28.6%)
Score: 2	0	1 (33.3%)	16 (38.1%)	0	7 (26.9%)	2 (50.0%)	8 (50.0%)	4 (57.1%)
Score: 3	0	0	5 (11.9%)	3 (42.9%)	4 (15.4%)	1 (25.0%)	2 (12.5%)	0
Modified Centor score: 0–1	12 * (100%)	2 (66.7%)	21 (50.0%)	4 (57.1%)	15 (57.7%)	1 (25.0%)	6 (37.5%)	3 (42.9%)
Modified Centor score: 2–3	0	1 (33.3%)	21 (50.0%)	3 (42.9%)	11 (42.9%)	3 (75.0%)	10 (62.5%)	4 (57.1%)
Initial antibiotics	11 (91.7%)	2 (66.7%)	41 (97.6%) *	6 (85.7%)	23 (88.5%)	3 (75.0%)	11 (68.8%) *	7 (100%)
Antibiotics by ID Dr.	8 (66.7%)	2 (66.7%)	40 (95.2%) *	2 (28.6%) *	23 (88.5%)	3 (75.0%)	7 (43.8%) *	5 (71.4%)

* Star symbol indicates *p* < 0.05 according to non-parametric test and chi-square test. Abbreviations: HSV—herpes simplex virus; EBV—Epstein–Barr virus; CMV—cytomegalovirus; ADV—adenovirus; HIV—human immunodeficiency virus; GAS—Group A Streptoccoci; VZV—varicella-zoster virus.

**Table 4 microorganisms-13-00628-t004:** Characteristics of different causal pathogens of acute pharyngotonsillitis.

FLU	HSV	EBV CMV	ADV	Enterovirus	HIV
Older age Cough Sudden onset of high fever, headache, and myalgia	Higher WBC/PMN, CRP (bacterial infection-like) Vesicles/ulcers Prolonged fever Tonsil exudate	Younger age FUO or prolonged fever Neck LAP/splenomegaly Atypical lymphocytes Hepatitis Tonsil exudate	Higher WBC/PMN, CRP (bacterial infection-like) Conjunctivitis Prolonged fever Tonsil exudate	Higher WBC/PMN, CRP (bacterial infection-like) Vesicles/ulcers Tonsil exudate	Younger age/male gender FUO or prolonged fever Neck LAP/splenomegaly Atypical lymphocytes/low WBC Hepatitis Rash BWL

Abbreviations: FLU—influenza; HSV—herpes simplex virus; EBV—Epstein–Barr virus; CMV—cytomegalovirus; ADV—adenovirus; HIV—human immunodeficiency virus; WBC—white blood cell count; PMN—polymorphonuclear neutrophil; CRP—C-reactive protein; FUO—fever of unknown origin; LAP—lymphadenopathy; BWL—body weight loss.

## Data Availability

The original contributions presented in this study are included in the article. Further inquiries can be directed to the corresponding author.

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
