# Peer review of "Do Hospitalized Adult Patients with Acute Pharyngotonsillitis Need Empiric Antibiotics? The Impact on Antimicrobial Stewardship"

_microorganisms, 2025, doi:10.3390/microorganisms13030628_

Round 1

Reviewer 1 Report

Comments and Suggestions for Authors

Dear Authors,

This article addresses the issue of antibiotic overuse in acute pharyngotonsillitis in hospitalized adult patients. It points out the unnecessary use of antibiotics, especially with regard to viral infections. It provides valuable data on the etiology of acute pharyngotonsillitis, showing that infections caused by GAS are in the minority and the majority are due to viral infections. This study points out that overuse of antibiotics can lead to antimicrobial resistance.

Major1: Perhaps the novelty of this paper is that 1) it is an adult patient, not a pediatric patient, and 2) it focuses on inpatients, not outpatients. However, neither the abstract nor the inclusion emphasizes this point.

Major2: The proposed diagnostic algorithm using the Modified Centor Criteria is expected to be applicable to clinical practice. However, please discuss more specifically in which patients and when it should be used in order to increase its clinical value.

Major3: 

The selection of patients for admission based on clinical criteria such as fever, severe sore throat, dysphagia, leukocytosis, and high C-reactive protein levels is commendable as a practically useful indicator in clinical practice, but it does not provide reference values or scoring for each item, for example, how many or more WBC or CRP were admitted? However, there is no reference value or scoring for each item.

Major4: Regarding the exclusion criteria, it is stated that “patients with a positive rapid influenza antigen test were excluded,” but while the rationale for excluding patients receiving additional antibiotic therapy to prevent serious complications associated with influenza (e.g., secondary bacterial pneumonia) is briefly explained, the exclusion criteria for patients with other viruses (e.g., Epstein-Barr virus and cytomegalovirus), exclusion criteria for those infected with these viruses are not stated.

Best regards,

Reviewer 2 Report

Comments and Suggestions for Authors

The authors presented here the report of 10-year surveillance data of hospitalized adult patients with acute pharyngotonsillitis in hospital, demonstrated the over-use of antibiotic in these hospitalized patients, which is leading the emerging of antibiotic resistance crisis globally. The data presented here is a great proof of unnecessary over-use of antibiotics, and can further draw the attention of novel determination methods and treatment of infections over the community and hospital, thus this paper is recommended to be published in the present form.

Round 2

Reviewer 1 Report

Comments and Suggestions for Authors

Dear Authors,

Influenza virus infection is omitted as a criterion for exclusion diagnosis. However, this paper also has epidemiological value for the overall acute tonsillitis in adults. Therefore, the number of cases excluded due to influenza virus should be reexamined and added to the text and Figure 1.

Best regards,

Dr. 

Author Response

Comments 1: Influenza virus infection is omitted as a criterion for exclusion diagnosis. However, this paper also has epidemiological value for the overall acute tonsillitis in adults. Therefore, the number of cases excluded due to influenza virus should be reexamined and added to the text and Figure 1.

Response 1: Thanks for your comments. We added the number of influenza patients excluded from our cohort in the text and Figure 1.